# GRAD-T: Graph Regularized Attention-based Diffusion Model for Analysis of Contextual Emotion Contagion

## Abstract

We propose a Graph Regularized-Attention-Based Diffusion Transformer (GRAD-T) model, which uses kernel temporal attention and a regularized sparse graph method to analyze model general diffusion processes over networks. The proposed model uses the spatiotemporal nature of data generated from diffusion processes over networks to examine phenomena that vary across different locations and time, such as disease outbreaks, climate patterns, ecological changes, information flows, news contagion, transportation flows or information and sentiment contagion over social networks. The kernel attention models the temporal dependence of diffusion processes within locations, and the regularized spatial attention mechanism accounts for the spatial diffusion process. The proposed regularization using a combination of penalized matrix estimation and a resampling approach helps in modeling high-dimensional data from large graphical networks, and identify the dominant diffusion pathways. We use the model to predict how emotions spread across sparse networks. We applied our model to a unique dataset of COVID-19 tweets that we curated, spanning March to December 2020 across various U.S. locations. We used model parameters (attention measures) to create indices for comparing emotion diffusion potential within and between nodes. Our findings show that negative emotions like fear, anger, and disgust demonstrate substantial potential for temporal and spatial diffusion. We will release the dataset for public consumption. Using the dataset and the proposed method we demonstrate that different types of emotions exhibit different patters of temporal and spatial diffusion. We show that our model improves prediction accuracy of emotion diffusion over social medial networks over standard models such as LSTM and CNN methods. Our key contribution is the regularized graph transformer using a penalty and a resampling approach to enhance the robustness, interpretability, and scalability of sparse graph learning.

## 1 Introduction

Graph learning provides a versatile and unified framework for modeling diverse real-world scenarios where the underlying data generation process involves interconnected entities such as infection transmission, emotion contagion, traffic flows, and knowledge dissemination. Several existing neural network-based models such as Graph Neural Networks (GNN), (Dwivedi and Bresson, 2020; Tsitsulin et al., 2023), and Graph Convolution Networks (GCN) (Wu et al., 2023b). suffer from scalability issues for large graphs, over-smoothing, and the inability to capture complex dependencies (Gao et al., 2023). Graph Transformers overcome these limitations by using an attention mechanism to capture dependencies across nodes in a graph. This method avoids relying on local aggregation and reduces the risk of over-smoothing (Hussain et al., 2022; Mialon et al., 2021; Zhang et al., 2020; Ma et al., 2022). Additionally, emotion diffusion over social and physical networks emotions plays a fundamental role in human behaviors such as interpersonal relationships, political affiliations, and commercial transactions. A key characteristic of emotions is that an individual or a group can influence the emotions of other individuals or groups, a process that is termed as "emotion contagion" (Elfenbein, 2014). Emotions are "the generated states in humans that reflect evaluative judgments of the environment, the self and other social agents"(Hudlicka, 2011). However, the detection of contagion networks is challenging because the influence networks on social media are often latent and not directly observed, and

suffer from the "curse of dimensionality" due to the large number of influence pathways (Cabannes et al., 2021). For a network graph with $n$ vertices, there are $n^2$ distinct edges and $2^{n^2} - 1$ pathways over which emotion diffusion (contagion) can occur. However, Not all these pathways are important or interesting, instead only a few pathways are of the essence. Extant graph transformer models do not explicitly consider the sparsity of influence structure in a diffusion network. The COVID-19 pandemic highlighted how emotions shared on social media affected public responses to measures like lockdowns and mask-wearing, influencing the virus's spread (Tsao et al., 2021). While the COVID-19 pandemic is a few years past, we can still learn from COVID-19 for the future.

In this paper, we address the above challenges by proposing the Graph Regularized-Attention-Based Diffusion Transformer (GRAD-T), a regularized graphical transformer model that incorporates both temporal and spatial diffusion of emotions over sparse networks. In our proposed model, we use a Kernel attention mechanism for temporal attention, and softmax spatial attention. We regularize the attention matrix following the Graphical LASSO (G-LASSO) structure (Cheng et al., 2014) to identify the predominant influence structures. Furthermore, for computational ease, we employ a sampling-based block-diagonal regularization of the graph attention. Graphical data analysis has been used in several contexts that involve data generated from a network of connected nodes over a graph. Sparse estimation of graphical data involves estimating the minimizer of the penalized positive definite matrix $\Theta$ such that $\hat{\Theta} = \arg\max -\mathcal{L}(\Theta) + \lambda\|\Theta\|$, where $\mathcal{L}(\Theta)$ is the log-likelihood of graphical data of the form $n\left(\log\det\Theta - Tr(\frac{1}{n}XX^T\Theta)\right)$, where $X$ is the observed data (Hallac et al., 2017). The Representer Theorem (Schölkopf et al., 2001; Romero et al., 2016) shows that the solution to the regularization problem stated here can be represented as $f(v) = \sum_{s=1}^{n} \alpha_s k(x_s, x_v)$, where $x_v$ is a focal node data, $\alpha_s$ are some affine constants, and $k(\cdot)$ is a suitable Kernel function. We use the Representer theorem to model the temporal dynamics within location using a kernel learning process. Kernel transformation is akin to converting information from $X_m$, representing $n$ individuals, into a symmetric kernel $K_m(\cdot)$ to capture the population distribution as similarities among samples. Usually it is computationally hard to consistently estimate regularizers of large graphs due to curse of dimensionality. We combine kernel learning with a repeated sub-sampling process to estimate the regularized graphical spatial attentions. The regularization process helps control model complexity, preventing over-fitting, and enhances the model's generalizability and interpretability by highlighting important diffusion pathways (Wang et al., 2023). Additionally, we integrate topic encoding into the model, which enhances the contextual relevance of the analysis (Wang et al., 2020). Finally, using the proposed model we develop a set of indices to measure and compare the diffusion potential of different emotion types. We use these indices to demonstrate that different emotion types exhibit different levels of diffusion potential over spatial and temporal dimensions. These indices help us quantify the diffusion potential of different emotion types and different source nodes. Specifically, we show that negative emotions are sustained over a relatively longer time-period. Similarly, negative emotions, as compared to positive emotions, spatially diffuse to a larger extent. These findings are important in many contexts and help us understand the differential herding of human emotions and sentiments in many different contexts, such as infectious disease transmission, financial market trends, and consumer sentiments and ratings of products and services.

We collected a unique dataset of COVID-19-related tweets containing 10 million unique tweets that were retweeted 40 million times from April 1, 2020, to August 31, 2020, covering 879 zip codes in the US. The tweets focus on topics like mask mandates, lockdowns, and testing. The COVID-19 pandemic highlighted how emotions shared on social media affected public responses to measures like lockdowns and mask-wearing, influencing the virus's spread (Tsao et al., 2021). While the COVID-19 pandemic is a few years past, we can now learn a lot more from COVID-19 for the future. Figure 1 illustrates emotion contagion across four locations. The individuals represent social media users who express emotions such as "joy" or "anger", depicted by green and red circles, respectively. For example, on Day 1, a user at Location 1 expresses "joy," while another at Location 2 expresses "anger". By Day 2, more users express these emotions, influenced by prior opinions from the same or different locations. The spread of emotions depends on the similarity (affinity) of locations and users, shaped by dominant social beliefs and norms.

Finally, our work contributes in the following manner:

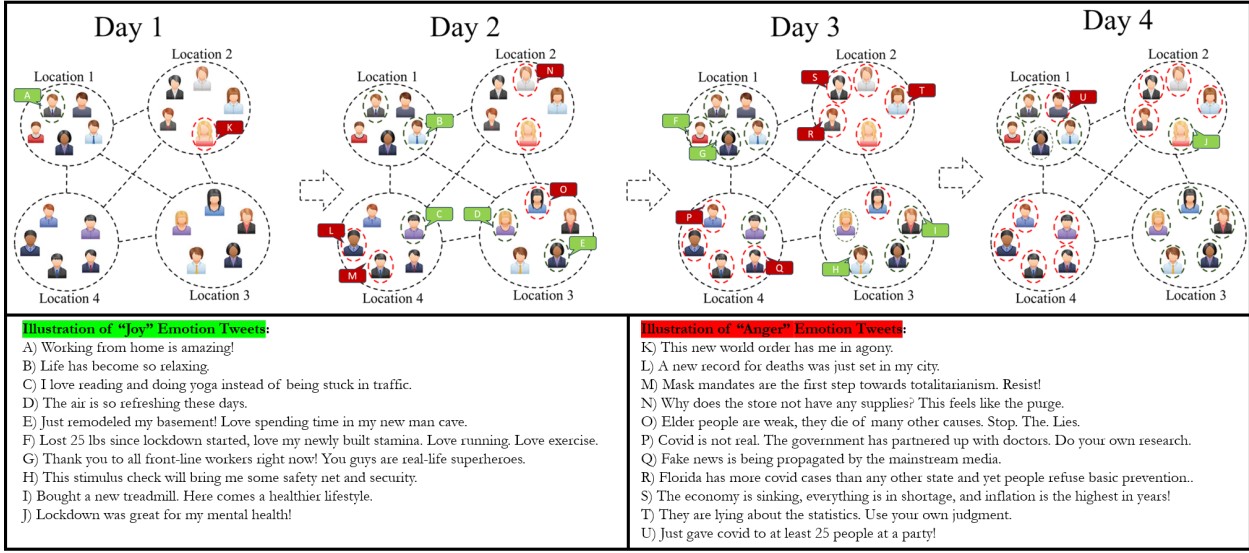

Figure 1: Illustration of the Data Generation Process that we Model in GRAD-T. We consider four locations and four time points to demonstrate the diffusion of emotions across time and spatial dimension. Consider that on day 1 a tweet (Tweet A) from location 1 denotes an emotion of "Joy" and another tweet (Tweet K) from location 2 denotes "Anger". These tweets can influence the expression of similar emotions in subsequent posts from the same or different locations, for example, Tweets B, C, D, and E denote "Joy", and Tweets L, M, N, and O denote "Anger". The influence depends on the Affinity of different locations, however, this Affinity structure is considered to be unobserved.

- We propose a Graph Regularized-Attention-Based Diffusion Transformer (GRAD-T) model, which uses kernel temporal attention and a regularized sparse graph method to analyze how emotions spread across sparse networks.

- We propose two different regularizations, namely, purely data-driven regularization based on $L_1$ penalty, and a sampling-based regularization.

- The proposed model is focused on sparse learning of latent graph structures, which can highlight dominant diffusion pathways, thus augmenting interpretability.

- We develop several summary indices to measure diffusion potential on graphs.

- We curated a unique dataset based on COVID-19 tweets, which will be shared for public research.

- Finally, we use the proposed model and the COVID-19 dataset to demonstrate that negative emotions such as fear and anger tend to exhibit relatively longer temporal and wider spatial diffusion potentials.

## 2 Related Work

### 2.1 Emotion Detection using Transformers

Current state-of-the-art literature on emotion detection uses BERT-based models (Yang et al., 2019) such as ALBERT (Lan et al., 2019), or SpanBERT (Joshi et al., 2020) to encode the text embedding, which is then passed on to a classifier to estimate the emotion weights. Several more subsequent papers used Transformer architecture to classify emotions (Zanwar et al., 2022; Palani et al., 2021; Xu et al., 2019). Most of these models use pre-classified dictionaries to encode emotions in text data. In contrast, we also incorporate the topical context of emotional states to improve the interpretability of model outcomes.

## 2.2 Graph Diffusion using Transformers

Early work on graph-based neural networks proposed recursive networks (Scarselli et al., 2018) or convolutional networks (Kipf and Welling, 2016). Recent literature has expanded transformer architecture to modeling graph structures. Dwivedi and Bresson (2020) proposed a graph-based transformer model. Several other papers have used Graph Laplacian or an affinity matrix as the positional encoding for graph transformer modeling (Lim et al., 2022; Beaini et al., 2021; Wang et al., 2022). In contrast, recent papers used a graph intimacy matrix that indicates the intimacy or affinity score between pairs of nodes (Shi et al., 2022). Graph transformers have been used for modeling sequence learning (Cai and Lam, 2020), molecular structure modeling (Wu et al., 2023a), full document modeling (Nguyen et al., 2021), image classification, cross-language modeling (Ahmad et al., 2021), and 3D model generation (Lv et al., 2024). All these models require specifying the graph adjacency or structure. We extend the graph diffusion models using transformer architecture to contexts where the influence structure is not explicitly observed, and the influence is temporal, i.e., varies over time. We incorporate a Kernel encoding mechanism to model the spatial and temporal influence and affinity within and across graph nodes.

# 3 The Proposed Model: GRAD-T

In this section, we develop the proposed model. First, we describe the problem, describe a conceptual diffusion framework, and finally, we develop the proposed model based on the diffusion framework in the following three sections.

## 3.1 Problem Description

Let $W_{i,j,t}$ denote a tweet from user $i$ at location (a city or zip code) $j$ at time period (day or week) $t$. Let $E_{i,j,t} \in \mathcal{E}$ denote the $d$-dimensional emotion vector $\{e_{i,j,t}^1, \ldots, e_{i,j,t}^d\}$, where $e_{i,j,t}^k$ denotes the probability that the tweet $W_{i,j,t}$ belongs to the emotion dimension $k$. Also, let $Y_{i,j,t} \in \mathcal{Y}$ denote a $p$-dimensional topic vector $\{y_{i,j,t}^1, \ldots, y_{i,j,t}^p\}$ embedded in the tweet $W_{i,j,t}$, where $y_{i,j,t}^k$ denote the probability that the tweet is about topic $k$. The matrix $E_{i,j,t}^Y = E_{i,j,t} Y_{i,j,t}^T \in \mathcal{E} \times \mathcal{Y}$ is a $d \times p$ dimensional matrix, where each column denotes the topic-specific emotion vector in the tweet $w_{i,j,t}$. Since each sender ID is not repeated sufficiently in the dataset, we consider the location-specific topical emotion matrix by averaging the emotion matrices of all IDs from a location $j$ at time $t$, i.e., $E_{j,t}^Y = \frac{1}{|W_{j,t}|} \sum_{w_{i,j,t} \in W_{j,t}} E_{i,j,t} Y_{i,j,t}^T$. $G = (V, E)$ denotes a graph, where $V$ is the set of locations with $|V| = n$, and $E$ denotes the latent directed edge set (affinity structure). The objective of the paper is to predict the future topical emotional state of a location, given the past emotions, and the latent dependence (affinity) graph across locations. During COVID-19, which we use as the context for the numerical analysis here, different locations (cities or zip codes) exhibited different emotional states related to vaccines, masks, and other preventative measures. These location-specific population-level emotional states specific to different topical contexts are important to analyze and predict for several purposes including prediction of disease spreads during epidemics, and implementation of preventative measures. Therefore, the specific problem that we propose to model here is to learn a function $\phi : G, E_{i,1 \ldots t}^Y, E_{j,1 \ldots t, j \in \{1, \ldots, n, i \neq j\}}^Y \to E_{i,t+1}^Y$. The objective is to estimate a machine-learning model that can predict the future topical emotion state with precision and consistency.

## 3.2 Graph Diffusion Framework

Emotions manifested in Social Media expressions represent the cultural and social norms, political beliefs, and public opinions about matters such as political decisions, social issues, or natural disasters. These emotions such as joy, anger, happiness, and sadness present the belief polarities of the population about public events and matters. These beliefs and norms that determine the expressed emotions form and change over time based on the history of such beliefs and norms within a node, and based on the influence of similar beliefs and norms that other connected nodes exert on the focal node through direct or indirect interactions. Therefore, To develop the graph transformer-based diffusion model, we consider that the expressed emotions

in a node of the graph are influenced by the history of the emotions within a node, and the expressed emotions of other connected nodes as expressed by the following diffusion equation 1.

In this section, we develop a mathematical model of the diffusion process as in Equation 1,

$$
\begin{aligned}
\frac{d\boldsymbol{E}_t}{dt} &= \boldsymbol{\beta}' \boldsymbol{E}^p - \gamma L \boldsymbol{E}_t \\
&+ \boldsymbol{\eta}_t : \boldsymbol{E}^p = \{\boldsymbol{E}_t - \boldsymbol{E}_{t-1}, \dots, \boldsymbol{E}_t - \boldsymbol{E}_{t-p}\},
\end{aligned}
\tag{1}
$$

where, $\boldsymbol{\beta} = \{\beta_1, \dots, \beta_k\}$ is the auto-regressive coefficient, $\gamma$ is the spatial diffusion coefficient, $A$ is the adjacency matrix, $L = D - A$ is the Laplacian of the graph with $D_{i,i} = \sum_{j=1}^{n} A_{i,j}$ being the degree matrix, and $\boldsymbol{E}_t = \{E_{1,t}, \dots, E_{n,t}\}$. The additive separation of the within-location temporal dependence and the across-location spatial dependence is convenient from the perspective of ease of estimation and avoidance of over-fitting, particularly in contexts characterized by sparse signals. We generalize the above diffusion model in Equation 2,

$$
\frac{d\boldsymbol{E}_t}{dt} = \sum_{k=1}^{p} K_h(\boldsymbol{E}_{t-k}^p) + \sum_{j=1}^{n} f\left(L, \boldsymbol{E}_{j,t}\right) + \boldsymbol{\eta}_t,
\tag{2}
$$

where, $K_h(\cdot)$ is a kernel temporal attention function with tuning parameter $h$, and $f(\cdot)$ is a spatial attention function that captures the across-node dependence in a graph. The Equation 2 forms the basis for the graph diffusion model.

### 3.3 Proposed Model Details: GRAD-T

Figure 2 shows the structure of the proposed Graph Diffusion model. The model incorporates within-node temporal dependence (temporal kernel attention) and across-node spatial dependence (softmax spatial attention) as per Equation 2. The key innovation in the proposed model is in estimating a regularized sparse attention matrix in two different ways, which allows us to estimate the dominant spatial dependence structure in a latent graph. The prediction layer (GNN) uses the sparse attention matrix as input to predict the output, the emotion states for the next period. Below, we describe the input embedding, the input graph structure, the temporal attention mechanism, the spatial attention mechanism with regularized sparse attention matrix computation in two different ways, and the prediction layers.

#### 3.3.1 Emotion Classification

The node features consist of the topic-specific emotion vectors. Therefore, the estimation of the emotion vector is an important pre-processing step. While the specific emotion detection model can be replaced by other models, we used a BERT model for the emotion vector generation for each node $i$ at time $t$ Delbrouck et al. (2020); Hazmoune and Bougamouza (2024). The BERT-based emotion estimation uses one hot-coded word sequence and positional encoding for probabilistic emotion classification. The output $\boldsymbol{E}_{i,t}$ is a vector of weights for all emotion dimensions.

#### 3.3.2 Topic Specific Emotions

Topic modeling from a corpus of text documents such as tweets has played an important role in text analysis. Several methods of topic modeling have been proposed including Latent Dirichlet Allocation (LDA), and Embedded Topic Models (ETMs). Given the spirit of the work, we used a BERT-based topic model in line with extant work Basmatkar and Maurya (2022); Reuter et al. (2024). Again, the specific method used is not very important. The output of the topic modeling is denoted as $Y_{i,t}$, which is the vector of probabilities of a tweet belonging to a vector of topics. We create the topic-specific emotion matrix as $E_{i,t}^Y = E_{i,t} Y_{i,t}'$. The final node feature embedding for our graph diffusion model is a vectorized version of the topic-specific emotion matrix denoted as $vec(E_{i,t}^Y)$.

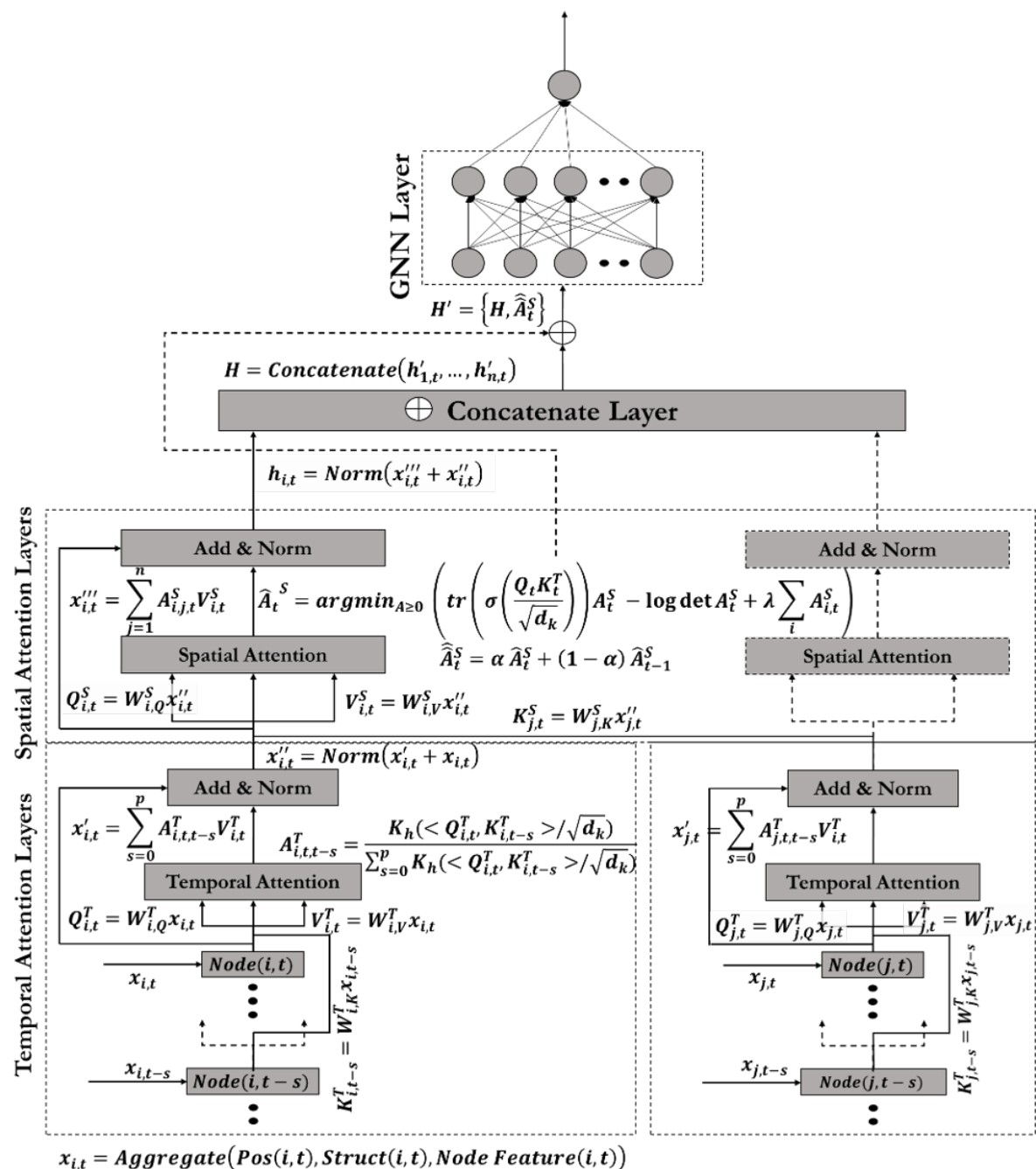

Figure 2: **GRAD-T**: Graph Regularized-Attention-Based Diffusion Transformer with Temporal Kernel Attention within Location $i$, and Regularized Spatial Attention from Other Locations $j \in \{1, \ldots, n; i \neq j\}$.

### 3.3.3 Positional Embedding

In this section, we describe the graph embeddings, which form the central elements of our proposed models. In a graph diffusion transformer, positional embedding encodes the structural information about a node's position within the graph. This can be achieved through various methods, including absolute positional embedding, which uses centrality measures; relative positional embedding, based on shortest paths or distances from other nodes; spectral embedding, derived from the eigenvalues and eigenvectors of the graph Laplacian; and random walk embedding, which utilizes random walk distances from a node. However, all these embeddings require a structured graph as input. In our context like many other contexts, the graph structure is latent and we do not directly observe the graph structure. We follow two different strategies to learn the graph structure. First, we use a dense fully-connected graph as input and use a regularization process at the spatial attention matrix estimation step to learn a sparse structure. Second, we propose a sampling technique with side information to learn a prior graph structure, which is used as input for the subsequent analysis. Let us consider the eigenvectors of the graph $G$ to be $\{u_1, \ldots, u_n\}$. The eigenvectors capture the graph's connectivity structure and represent the nodes' positional information with respect to the other nodes.

**Fully connected graph.** For a fully connected graph (with no prior structure information) of $n$ nodes, the first eigenvector is given by $u_1 = [1\ 1\ \ldots\ 1]$ with the corresponding eigenvalue $\lambda_1 = 0$. The subsequent eigenvectors are of the form $u_k = [-1, \ldots (k-1)\ times, -1, k, 0, \ldots, 0] : k \in \{2, \ldots, n\}$, and the corresponding eigenvalues are all equal to $n$. One limitation of this method is that for large graphs, the fully connected dense graph is not very informative and can lead to computational challenges.

**Graphs with side information.** Often there is side information that can be used to initialize the graph structure that is not as naive as the fully connected graph. In our context, "retweets" from other locations serve as a prior estimate of nodal connectivity. Additional side information, such as mobility measures, road and air connectivity, distances between locations, and social media metrics like friends and followers, can also inform prior graph structure. We use "retweet" information to construct the prior adjacency matrix. We compute the number of tweets of a location $i$ that has been retweeted at any point in time over the observation period with a small positive random Uniform distributed noise as the adjacency weight $w_{i,j}$, i.e., $w_{i,j} = \delta_{i,j} + \epsilon$, where $\delta_{i,j}$ is the number of tweets from location $j$ that has been retweeted by individuals from location $i$, and $\epsilon_i$ is a uniformly distributed random noise. The positional embeddings are computed by computing the eigenvectors of the weighted input graph.

### 3.3.4 Structural Embedding

The structural embedding captures the structural properties of a node in a graph. The structural embedding is computed as the number of nodes connected to a node in a fully connected input graph and as the sum of the edge weights of the un-normalized adjacency matrix as described above. Let the structural embedding be denoted by $s_k$ for node $k$.

### 3.3.5 Input Embedding

The input embedding for each node is a vector of embeddings computed above, namely, the positional embedding, structural embedding, and vectorized topic-specific emotion embedding as node features. Therefore, the input embedding for node $i \in \{1, \ldots, n\}$ is $x_{i,t} = [u_i\ s_i\ vec(E_{i,t}^Y)]$.

### 3.3.6 Temporal Attention

In this and the next section, we describe the attention mechanisms of our model (ref. Figure 2). Several earlier models have used different kernel attention for temporal modeling Tsai et al. (2019); Song et al. (2021). The purpose of the temporal attention mechanism is to incorporate the time dependence of emotions in tweets from past periods. The attention to a tweet emotion $e_{i,t}$ from a past period $e_{i,t-k}$ depends on the divergence of the tweet emotions, i.e., $\langle e_{i,t}, e_{i,t-k} \rangle$, as well as on the time gap $k$. We use a kernel function of the form

$K_h(e_{i,t}, e_{i,t-k}, k) = \phi_h(\langle e_{i,t}, e_{i,t-k}\rangle)) \times \phi_h(k)$ following standard literature on temporal kernel attention Tsai et al. (2019). The specific kernel function we use is an exponential kernel as in Equation 3.

$$A_{i,t,t-k}^T = \frac{\exp\{\frac{\langle Q_{i,t}^T, K_{i,t-k}^T\rangle}{h_1\sqrt{d_k}}\}\exp\{-\frac{k}{h_2}\}}{\sum_{k=0}^p \exp\{\frac{\langle Q_{i,t}^T, K_{i,t-k}^T\rangle}{h_1\sqrt{d_k}}\}\exp\{-\frac{k}{h_2}\}}V_{i,t}^T,$$

$$Q_{i,t}^T = W_{i,Q}^T x_{i,t}, \ K_{i,t-k}^T = W_{i,k}^T x_{i,t-k}, \ V_{i,t}^T = W_{i,V}^T x_{i,t}, \quad (3)$$

where, $(h_1, h_2)$ are two tuning parameters, and $(W_{i,Q}^T, K_{i,t-k}^T, W_{i,V}^T)$ are weight matrices that are learned from the data. The output of the temporal attention layers are $x_{i,t}'' = Norm(x_{i,t}' + x_{i,t})$, where $x_{i,t}' = \sum_{k=1}^p A_{i,t,t-s}^T V_{i,t}^T$.

### 3.3.7 Regularized Spatial Attention

The purpose of spatial attention is to capture the dependencies or affinities of different nodes on each other. The usual spatial attention matrix is expressed as $A_t^S = \sigma\left(\frac{\boldsymbol{Q}_t\boldsymbol{K}_t^T}{\sqrt{d_k}}\right) : Q_t = [Q_{1,t}^S \ldots Q_{n,t}^S], K_t = [K_{1,t}^S \ldots K_{n,t}^S], Q_{i,t}^S = W_{i,Q}^S x_{i,t}'', K_{i,t}^S = W_{i,K}^S x_{i,t}''$, where $\sigma$ is a softmax computed over the rows of a matrix. However, as discussed earlier, a dense attention matrix for a large graph can lead to overfitting and noise and does not allow for the identification of the dominant dependencies. Therefore, we propose to estimate a regularized sparse attention matrix $\hat{A}_t^S$ in line with the G-LASSO penalty structure, noting that the attention values are all positive, as in Equation 4.

$$\hat{A}_t^S = \underset{A_t^S \geq 0}{\arg\min}\left(tr\left(\sigma\left(\frac{\boldsymbol{Q}_t\boldsymbol{K}_t^T}{\sqrt{d_k}}\right)\right)A_t^S - \log\det A_t^S + \lambda\sum_i A_{i,t}^S\right),$$

$$\hat{\hat{A}}_t^S = \alpha\hat{A}_t^S + (1-\alpha)\hat{A}_{t-1}^S, \quad (4)$$

where, $\lambda$ is a penalty term estimated during the training stage to maximize prediction precision on a hold-out cross-validation set, and $\alpha$ is a smoothing constant used to ensure that the attention matrices do not deviate wildly across time. We refrain from proofs of the regularization since $L_1$ matrix regularizers have been well studied in literature Kakade et al. (2012). One limitation is that for very large graphs, the training process is resource-intensive and time-consuming. Therefore, we propose the following numerical block-diagonal regularizing process.

In a sparse graph, it is possible to cluster the nodes in a block-diagonal structure. Many diffusion phenomena such as infection transmission, rumor and news diffusion, and emotion diffusion exhibit spatial clustering. A block diagonal matrix can represent such clustering, technically sub-space clustering with overlaps. Several papers have studied subspace clustering using block-diagonal matrices of graphical structures Lu et al. (2018); Xia et al. (2022). In the block-diagonal representation, the adjacency matrix of the graph $A$ is structured such that each column (representing nodes) is assumed to belong to $k$ subspaces, each subspace containing nodes denoted by submatrices such that $\boldsymbol{A} = [A_1, \ldots, A_k]$ and the submatrices are ordered according to their membership. One property of the block diagonal matrix is that it can be expressed as a linear combination of submatrices such that $\boldsymbol{A} = \boldsymbol{AZ}$, where $\boldsymbol{Z}$ is the representation matrix containing blocks of 1's and 0's ordered in the structure of the underlying block diagonal matrix Lu et al. (2018). A block-diagonal matrix from a given matrix can be obtained by solving the following optimization problem $\min_{\boldsymbol{Z}} \& \|\boldsymbol{Z}\|_1; \ s.t., \ \boldsymbol{A} = \boldsymbol{AZ}, \ diag(\boldsymbol{Z}) = 0$. The optimization is NP-hard and computation-intensive for large matrices. We propose to use a sampling method to create the block-diagonal sparsity of the input adjacency matrix, BD GRAD-T. We create a matrix $M$ such that the matrix entry $M_{i,j}$ represents the number of times a tweet from $i$ is either retweeted by node $j$ over the entire observation sample. We also, add a small positive noise to each element to ensure that no nodes are completely disconnected a-priori. This is done in consideration of the fact that no prior knowledge is complete and there is censoring of the

observations. We normalize each row to add to unity such that each row represents a probability distribution. Therefore, $M'_{i,j} = Norm_i(M_{i,j} + \epsilon)$, where $\epsilon$ is a small positive random noise such as a uniform or beta noise. Then for each node, we randomly sample a set of nodes with probability $M'_{i,j} : j \neq i$, which we call the affinity set $S_i$ with the corresponding weights corresponding to each node $i$ following a threshold criterion $\sum_{j \in S_i} M'_{i,j} \leq \theta$. Also, we perform this above step multiple times and take an average to induce some uncertainty in the affinity set selection process. In most contexts, following the Pareto principle, a few nodes cover the majority of the connection signals. The final weight matrix formed from the affinity vectors represents the initial adjacency matrix that is used as input for computing the weighted adjacency matrix $A$. The algorithm of the sampling-based sparse attention matrix estimation is shown in Algorithm 1.

---

**Algorithm 1** Algorithm for Block-Diagonal Sparse Graph Adjacency Matrix

---

**Require:** Input data $\{Y_{i,j,t}\}$, vector of retweets of tweets of node $i$ by members of node $j$.
  **Compute:** $M$ such that $M_{i,j} = \sum_y \mathbb{I}_{\{y \in \bigcup_{t=1}^T Y_{i,j,t}\}} + \epsilon : \epsilon \sim Uniform(0,1)$.
  **while** $N \neq 0$ **do**
    **for** $i = 1 \ldots n$ **do**
      **for** $j = 1 \ldots n : i \neq j$ **do**
        Normalize rows: $M'_{i,j} = \frac{M_{i,j}}{\sum_{j \neq i} M_{i,j}}$.
      **end for**
    **end for**
    **for** $r = 1 \ldots k$ **do**
      **for** $i = 1 \ldots n$ **do**
        Initialize $S_i = \emptyset$
        **while** $\sum_{j \in S_i} M'_{i,j} \leq \theta(0.8)$ **do**
          $S_i$ = Sample $j$ with probability weights $M_{i,\cdot}$.
        **end while**
      **end for**
      Construct $A^r$ such that $A^r_{i,j} = M'_{i,j} \mathbb{I}_{\{j \in S_i\}}$.
    **end for**
    $A = Average(A^1, \ldots, A^k)$.
    Compute spatial attention $A^S_t = \sigma\left(\frac{\boldsymbol{Q}_t \boldsymbol{K}^T_t M}{\sqrt{d_k}}\right)$.
  **end while**

---

### 3.3.8 Prediction Layers: GNN

The final layers are the prediction layers with Graph Neural Networks (GNNs). We use the regularized graph network or the block-diagonal network as the input graph with the node-specific attention vectors as the node features to predict the node-specific feature vectors to estimate a GNN model.

### 3.4 Indices to Measure Diffusion Potential of Emotions

We leverage the attention mechanism of transformer models to develop indices to measure and compare the diffusion potential of different emotion types in different contexts and applications. In transformer models, the attention measures are the relative weights that one entity has on the state of another related entity. Therefore, the attention measures provide us with the opportunity to develop measures that can be utilized to understand and analyze the relative diffusion of different emotions over time and across spatial dimensions. Specifically, we propose the following indices. For the following discussion, as indicated in the previous section, $A_{i,j,t}$ indicates the attention that node $i$ receives from node $j$ at time $t$. Similarly, $A_{i,t,t'}$ indicates the temporal attention that time $t$ receives from a previous time $t'$.

1. **Temporal Diffusion Index** is the attention that previous time periods have on the focal time and is therefore measured as follows, $T(k) = \frac{A_{i,t,t-k}}{\sum_{j=1}^t A_{i,t,t-j}}$. Note that we have normalized the measures by the attention vector of all previous periods. Higher values of $T(k)$ indicate higher levels of influence.

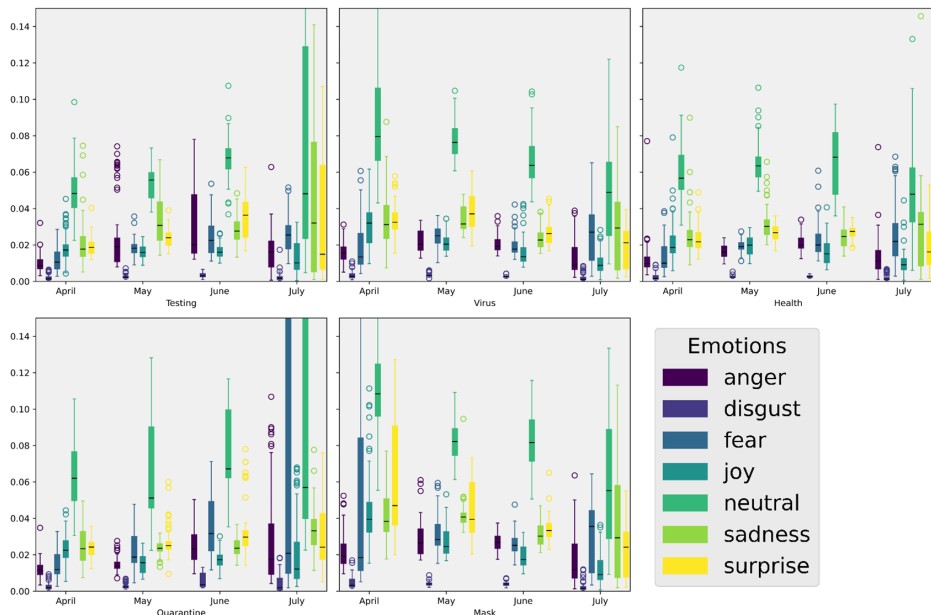

Figure 3: Temporal Distribution of Emotions Across Topics

2. **Inward Diffusion Index** is the attention that other connected nodes have on a focal node and is measured as follows, $I(i) = \frac{\sum_{k=1:i\neq j}^{n} A_{i,k,t}}{\sum_{k=1:i\neq j}^{n} A_{i,k,t} + \sum_{s=1}^{(t-1)} A_{i,t,s}}$.

3. **Outward Diffusion Index** is the attention that a focal node has on all other connected nodes and is measured as follows. $O(i) = \frac{\sum_{k=1:i\neq j}^{n} A_{k,i,t}}{\sum_{k=1:i\neq j}^{n} A_{k,i,t} + \sum_{k=1:k\neq i}^{n} \sum_{s=1}^{(t-1)} A_{k,t,s}}$.

# 4 Experiments

## 4.1 Dataset and Data Organization

The dataset is a unique proprietary dataset that was collected by the authors during the COVID-19 pandemic. The dataset was collected from Twitter (currently X) using COVID-19-related keywords such as *SARS-CoV-2, COVID-19, pandemic, mask, lockdown, infection, social distancing, etc.* A total of 89 different keywords were used for collecting Twitter posts daily from 879 different zip-codes of the united states covering 50 states. The Tweets were collected daily continuously for a period of 5 months starting April 01, 2020 going up to August 31, 2020. The original dataset, comprising 21 GB of text data from Twitter, was carefully processed. Python code was used to filter Tweets related to COVID-19, retaining only English-language Tweets for easier analysis. Approximately 50% of the Tweets were kept, resulting in about 10 million Tweets retweeted around 40 million times across the US. The dataset includes fields such as Tweet ID, latitude, longitude, location name, user ID, and Tweet text. This publicly shared dataset can aid in testing machine learning methods and assessing the impact of public sentiment on epidemic dynamics, providing valuable insights for pandemic response and policy-making. For the experiment, we used data from different locations in California, USA for the four months of April 2020 - July 2020.

# 5 Estimation and Results

The analysis code was developed using the Transformer library of PyTorch. For the experiment, we aggregate the data daily for each city and town. The analysis could also be done at a user level, however, given a large number of users, and more than 95% of the user IDs had only less than a total of 5 Tweets. We used

| Model Stage | Loss | MSPE | MAPE |
|---|---|---|---|
| No Spatial Atten. NN | 0.0439 | 0.0342 | 0.1527 |
| GCN | 0.0133 | 0.0108 | 0.0172 |
| Fully Conn. BERT | 0.0098 | 0.0089 | 0.0149 |
| Regul. GRAD-T | 0.0076 | 0.0045 | 0.0129 |
| **BD GRAD-T** | **0.0031** | **0.0039** | **0.0119** |

Table 1: Results of Data Analysis Experiments for the following models: (i) Neural Network with no Spatial Attention (No Spatial Atten. NN), (ii) Graph Convolutional Neural Network (GCN), (iii) Fully Connected BERT (Fully Conn. BERT), (iV) Regularized GRAD-T (Regul. GRAD-T), and (v) Block-Diagonal GRAD-T (BD GRAD-T).

the standard Transformer emotion library from the Transformer package to classify the text into emotion dimensions, namely, *Anger, Disgust, Fear, Joy, Neutral, Sadness, Surprise*. The training parameters are as follows: $K$ input row dimension is 5, $n$, input column dimension (nodes) is 879, the learning rate is 0.01, Epochs is 100, recurrent, the number of iterations to use the input data is 7, dimension of hidden states is 64, heads, the number of attention head is 8, the number of encoder attention layers is 6, the number of decoder layer is 6, dimension of the GNN layer is 256, and the dropout rate is 0.1. Finally, we used a standard LLM model to derive the topics from the tweets. Total 5 topics was identified, namely, "*mask*" containing keywords like "masking", "wear", "inconvenient"; "*testing*" containing keywords like "test", "nasal", "pcr", "positivity"; "*lock-down*" containing words like "quarantine", "lockdown", "mobility"; "*infection*" with words like "coronavirus", "covid", "infectivity", "positive"; and "*hospitalization*" with words like "health", "severe", "icu". We also computed the topic-specific emotions. Figure 4 presents the month-wise trend of the distribution of the different emotions for the five topics. From Figure 4a we observe that the different emotions exhibit significant variation across the four months of 2020. Interestingly, related to masking anger was relatively lower at the beginning of the pandemic and fear was relatively higher. As time progressed anger increased and fear reduced over the months of April 2020 to July 2020. These relate well to the infection prevalence in the US. Furthermore, in Figure 5, we show the distribution of spatial attention within the three spatial clusters for the month of July 2020. In Figure 5a we show the violin plot of the distribution of the topic specific spatial attention across locations within the three clusters. The diagram shows that "anger" and "fear" was the predominant emotion attention overall across all location, particularly in San Francisco cluster. In Los Angeles and San Diego cluster the predominant emotion attention was that of "sadness" and "surprise". Figure 5b compares the emotion-specific attention across locations in a radar diagram. This diagram also confirms the variation of the emotion attentions across different locations. The diffusion potential of "anger" and "fear" emotions, indicated by the spatial attention distributions, was the highest in the San Francisco cluster. In contrast, the diffusion potential of "surprise" and "sadness" was the highest in San Diego cluster. Finally, in the Los Angeles cluster, the higherst diffusion potential was that of "sadness".

## 5.1 Evaluation Method

We compared the proposed models GRAD-T and BD GRAD-T with other methods such as GCN, NN with no spatial connection and fully-connected graph transformer. We compare the prediction performance using three metrics, (i) the value of the loss function improvement over different epochs, (ii) Mean Square Prediction Error (MSPE) defined as $MSPE = \frac{1}{n} \sum_{i=1}^{n} \|E_{i,t} - \hat{E}_{i,t}\|^2$, and (iii) Mean Absolute Prediction Error (MAPE) defined as follows: $MAPE = \frac{1}{n} \sum_{i=1}^{n} \|E_{i,t} - \hat{E}_{i,t}\|_1$.

## 5.2 Results

We use the first three months' data to estimate the models and predict the emotions for the fourth month for the entire state of California, USA. Table 1 shows the estimation loss, and the MAPE and MSPE measures for GRAD-T and BD GRAD-T models as compared to the three baseline models. The spatial attention mechanism proposed in the GRAD-T model significantly reduces the MAPE and MSPE. The MAPE and

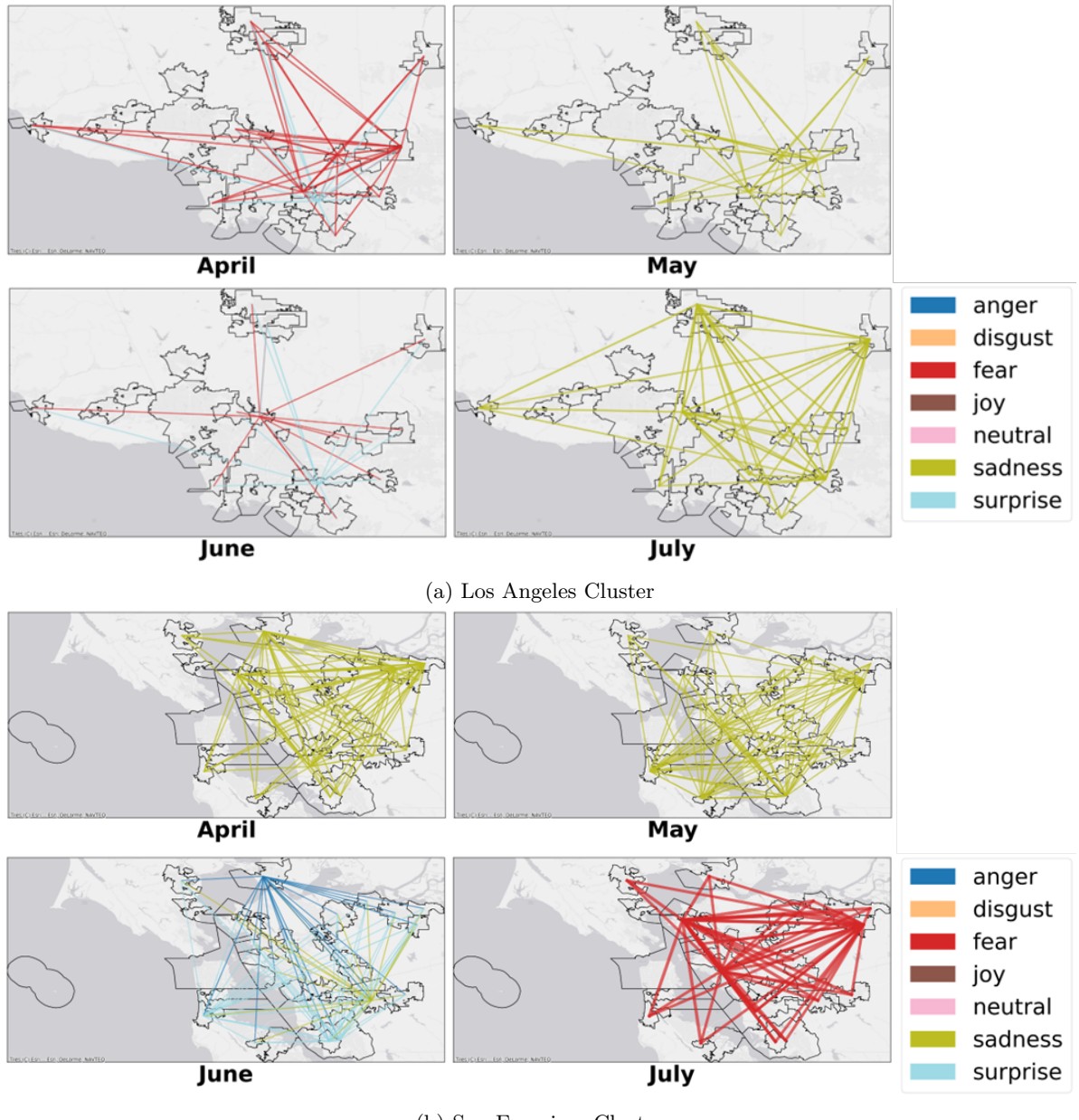

(a) Los Angeles Cluster

(b) San Francisco Cluster

Figure 4: Temporal Variation of Attention by Emotions

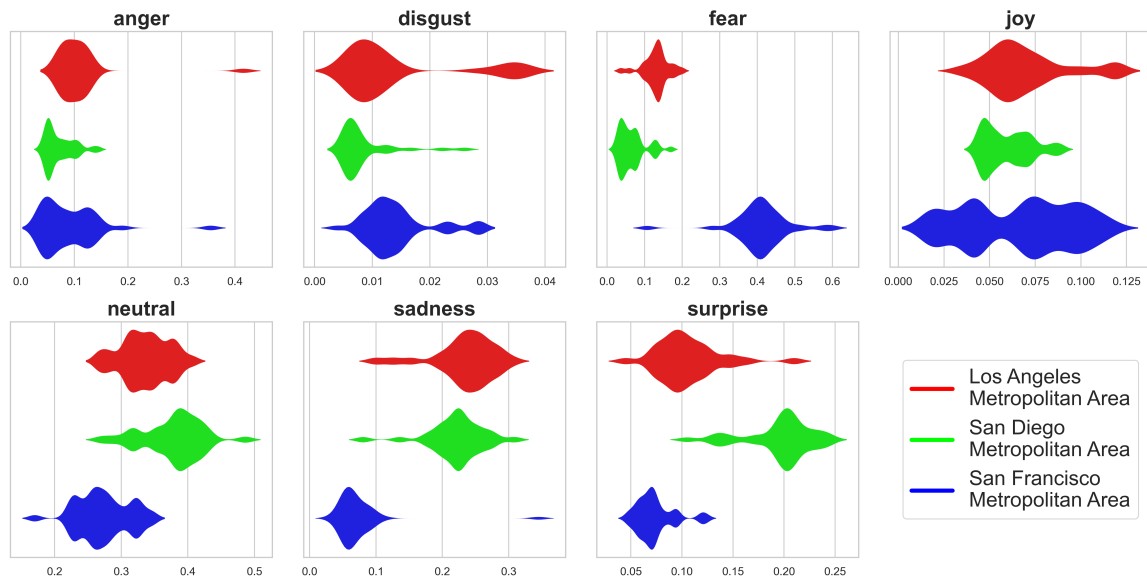

(a) Distribution of Spatial Attention within Different Clusters

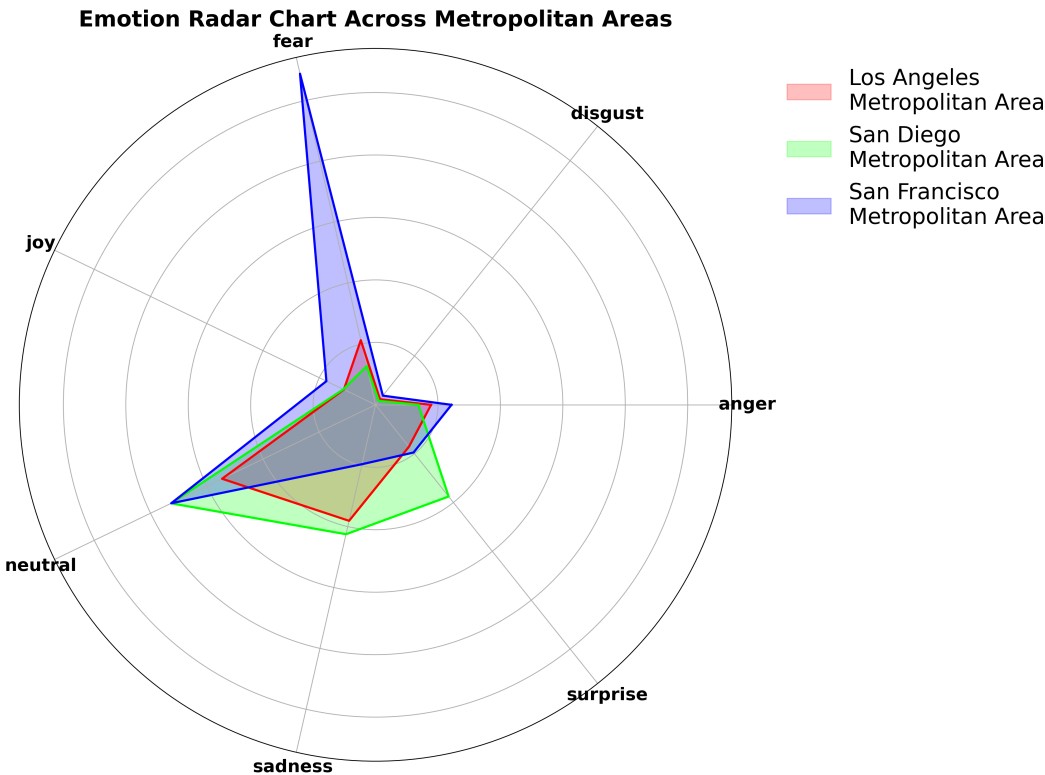

(b) Comparison of Emotion Specific Spatial Attention Across Different Clusters

Figure 5: Distribution and Variation of Spatial Attention Within Spatial Clusters

MSPE for the model with no spatial attention are 0.0342 and 0.1547 respectively. With the inclusion of spatial attention in the GRAD-T model, the measures respectively reduce to 0.0045 and 0.0129. The BD GRAD-T model further reduces the errors and performs the best in our experiment. This is because the inclusion of the retweet information adds to the performance of the model. The proposed models perform better than all the reference models. Next, we plot the attention values of two spatial sub-clusters, Los Angeles and San Francisco regions for illustration in Figures 4 and 6. In these figures, the color of the connections shows the emotion type, and the width of the connections shows the magnitude of the spatial attention. In the Los Angeles clusters, notice that initially in April, the dominant attention was that of fear. Indeed, the COVID-19 pandemic in California initially started in Los Angeles, and regions around the region were most affected. However, gradually the dominant attention by July was that of Sadness. This is understandable because the mortality from the COVID-19 pandemic peaked around July. In San Francisco regions, the emotion started with sadness and transformed into anger over time. For the topical attention, for both regions, the dominant attention was that of masking. Indeed, initially, the major issue that was discussed in media as well as in social media platforms was that of masking and the extent of masking required. There were several discussions on the shortage of masks and PPEs. However, the dominant attention for Los Angeles regions soon moved to that of quarantine, lockdowns, and testing. Indeed, the Los Angeles region witnessed mass lockdowns and large-scale population-level testing. The San Francisco cluster started with attention for masking, however, moved to quarantine and lockdowns with time, albeit with a delay with respect to that of Los Angeles. These attention measures broadly conform to the disease dynamics of California, which was one of the first states in the United States, along with New York, to be severely affected by COVID-19. These estimates demonstrate the applicability of the proposed model, and shows that the model can reasonably estimate spatial diffusion of emotions and sentiments over a graphical network in a reasonable manner.

Finally, in Figure 7 we show illustrative examples of the indices proposed earlier. Specifically, we show the temporal, inward, and outward dependencies for the emotions anger, fear, surprise, joy, and sadness. From the figures, we find evidence that anger and fear show relatively longer temporal dependence and wider spatial dependence. Specifically, the two negative emotions, anger and fear tend to linger more and spread more widely as evidenced by the wider distribution of the two corresponding indices. The indices provide interpretability to the estimates and allow us to observe simple and summary properties of the graph diffusion process, which are otherwise hard to observe. These observations show that negative emotions tend to have relatively longer temporal and wider spatial dependencies. Also, the dependencies show much wider variability across locations and time. In contrast, the distributions of joy and surprise in Figure 7 are relatively more compact and fleeting.

The interesting contextual insights that the experiment generated are the following. First, the regularization process is able to elicit dominant influence pathways as shown in the maps plots. Second, we show that negative emotions sustain for a relatively much longer duration. Third, the spatial dependence of the negative emotions is relatively higher than the positive emotions.

## 6    Conclusion

In this paper, we propose the GRAD-T model for contextual emotion detection and transmission prediction and use a relevant and unique dataset to demonstrate the performance of the proposed model as compared to the base model. The proposed model is an important extension of emotion detection and transmission models using Transformers. The model can be used for several practical problems and contexts. Finally, we find evidence that negative emotions such as fear and anger tend to show longer temporal dependence and wider spatial dependence. The limitations and possible future extension of the model are: (i) We do not include edge features such as distance matrices in the model, which can add predictive power, (ii) there are other regularization methods such as Bayesian regularizers that we have not explored, and (iii) we eliminated lots of estimation details due to space limitations. We will make those available upon request.

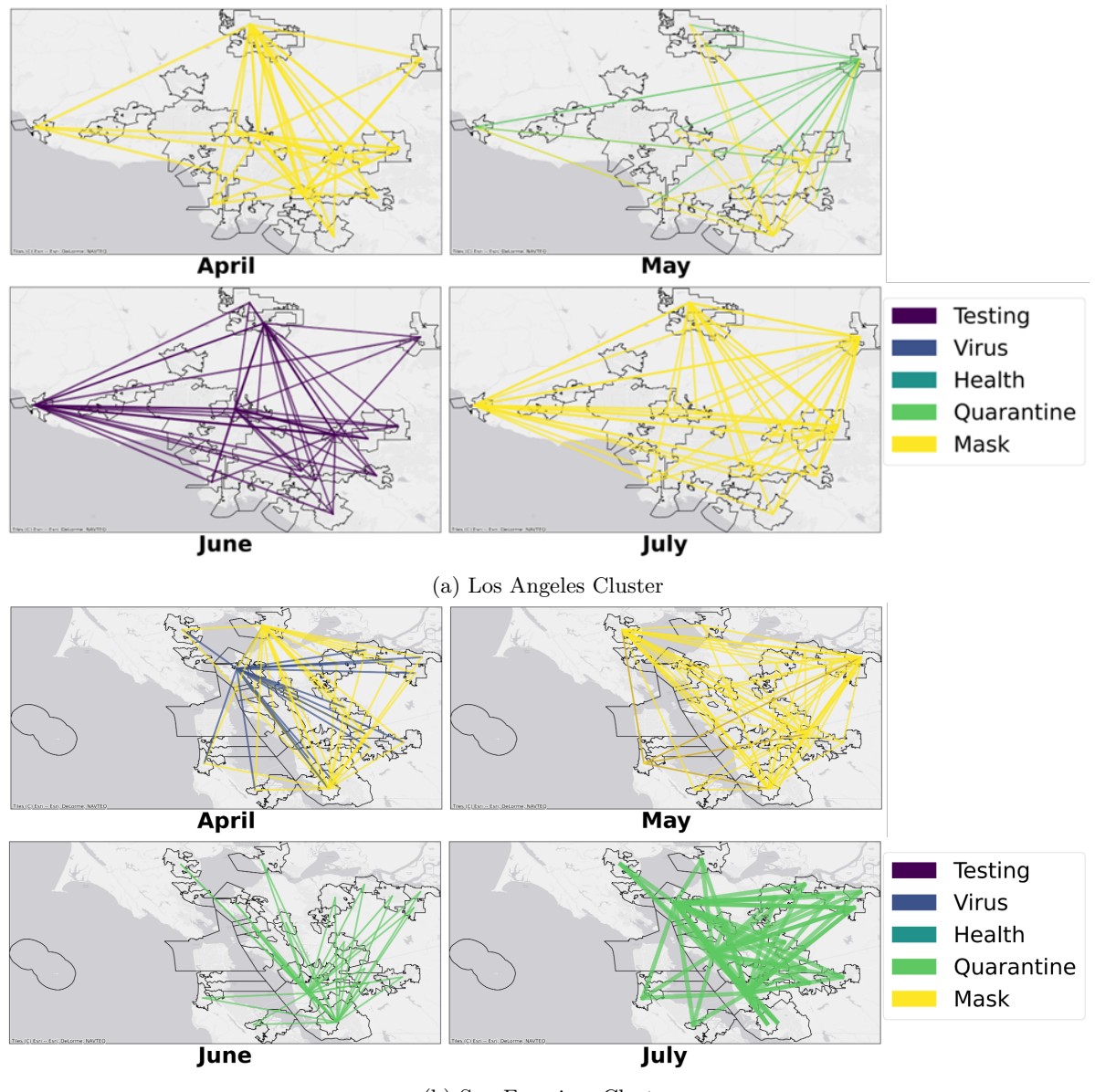

(a) Los Angeles Cluster

(b) San Francisco Cluster

Figure 6: Temporal Variation of Attention by Topic.

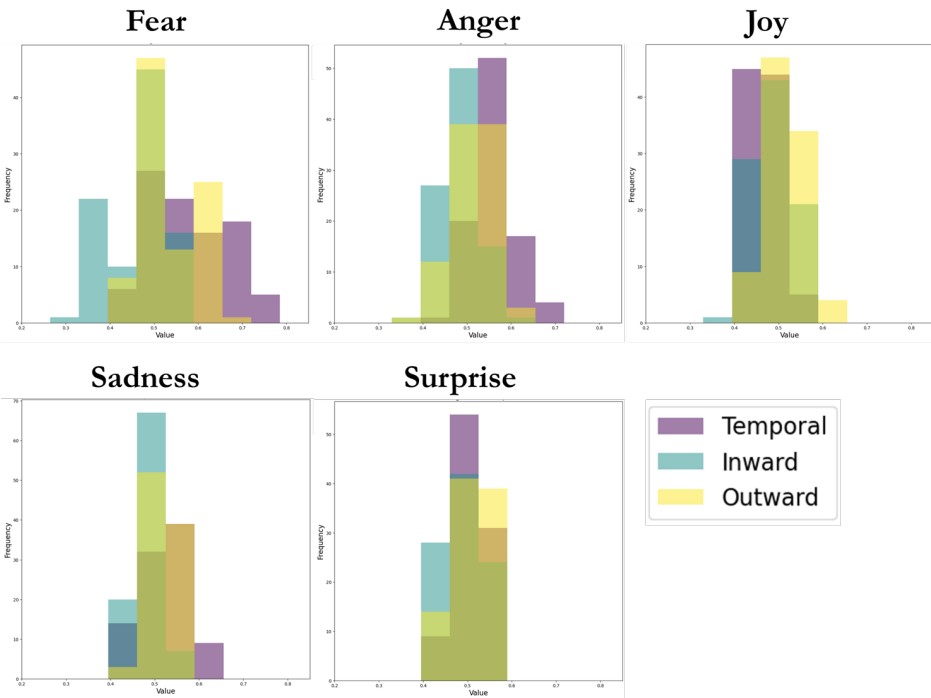

Figure 7: Illustrative Histograms of the Graph Indices

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
