# OpenReview forum: "GRAD-T: Graph Regularized Attention-based Diffusion Model for Analysis of Contextual Emotion Contagion"
_TMLR — Rejected by TMLR_

### Review · Reviewer_NVse · 2025-05-10

**Summary Of Contributions:**

This work proposes a novel attention based diffusion approach which uses kernel temporal attention for time dependence and uses regularized sparse attention for location dependence. It clearly specify position embeddings and two regularization methods for graph w.r.t locations. Moreover, it shows the analysis on the Covid datasets compared with other methods.

**Audience:**

Yes

**Claims And Evidence:**

Yes

**Requested Changes:**

1. Revise the notation

Formula/Notation issues:
(a) In the (3) A seems to be the score matrix and thus there should not have V^T there.
(b) x'_{i, t} looks wrong, that would be sum up with different k = 1 .. t not s.
(c) adding the dimension on (3) can make it clearer.
(d) for (4), can you replace sigma(QK^T/sqrt(d)) by A instead?
(e) Can you double check the dimension for the (4) in terms of the G-LASSO, it seems the min object is not a scalar.
(f) In figure 2, should not x''' = sum A_ijt V_j?
(g) notation needs improvement. For example, it uses both A_ijt and A_itt', which causes confusing.
(h) notation "p" may be abused, since p refers to both number of topics and the time lag order.

2. Add ablation study section.
3. Compare with stronger baseline results.

**Strengths And Weaknesses:**

Strengths:
This work provides a good literature review. And the methodology is easy to follow.


Weakness:
1. It have many formula/notation confusion.
2. It lacks of sufficient ablation study to justify those motivations such as kernel encoding, regularization, embeddings design.
3. The comparison in Table1 looks weak. Can author use stronger baselines for comparison?

Formula/Notation issues:
(a) In the (3) A seems to be the score matrix and thus there should not have V^T there.
(b) x'_{i, t} looks wrong, that would be sum up with different k = 1 .. t not s.
(c) adding the dimension on (3) can make it clearer.
(d) for (4), can you replace sigma(QK^T/sqrt(d)) by A instead?
(e) Can you double check the dimension for the (4) in terms of the G-LASSO, it seems the min object is not a scalar.
(f) In figure 2, should not x''' = sum A_ijt V_j?
(g) notation needs improvement. For example, it uses both A_ijt and A_itt', which causes confusing.
(h) notation "p" may be abused, since p refers to both number of topics and the time lag order.

---

### Review · Reviewer_nnM6 · 2025-05-22

**Summary Of Contributions:**

This paper introduces GRAD-T, a Graph Regularized Attention-based Diffusion Transformer model designed to capture the spatio-temporal dynamics of emotion contagion in networks. The approach combines a temporal kernel attention mechanism with a regularized spatial attention module, using either G-LASSO or a novel block-diagonal sampling strategy. The model is applied to a proprietary COVID-19 Twitter dataset spanning multiple U.S. locations. The authors also propose novel indices to quantify and interpret the temporal and spatial diffusion potentials of emotions. The model demonstrates improved predictive performance over several baselines, including GCN, neural networks without spatial connections, and fully connected graph transformers. Finally, the authors analyze the attention-derived diffusion indices.

**Audience:**

Yes

**Broader Impact Concerns:**

No ethical concerns.

**Claims And Evidence:**

No

**Requested Changes:**

**C1** - The clarity and formalism of the mathematical formulations must be significantly improved. All variables and equations should be clearly defined and derived with appropriate context.


**C2** - The authors should improve the presentation and justification of each model component, and establish clearer logical links between the different sections, particularly in Subsections 3.x.


**C3** - All baseline methods (GCN, fully connected BERT, NN without attention, etc.) must be introduced before the results are presented. The paper should also include implementation details such as hyperparameters and training procedures.


**Question** : The GRAD-T model and the proposed diffusion indices seem applicable beyond emotion contagion, or at least beyond this particular dataset. Have you tested your method on other public datasets to assess its generalizability and compare it against other state-of-the-art algorithms?

**Strengths And Weaknesses:**

**Strengths**

The main strength of this work lies in the creation of a new dataset of COVID-19-related tweets, specifically tailored for studying emotional contagion on social media. This dataset enables the use of advanced architectures such as graph transformers, and, if made publicly available, could significantly benefit the research community. Moreover, the effort to enhance interpretability - through the analysis of attention matrices and the introduction of diffusion indices - is especially relevant and provides valuable insights into the dynamics of the phenomenon studied.


**Weaknesses**

**W1**: There are several issues with mathematical formulation and notation. Key equations (e.g., Equations 1, 2, and 3) are introduced without proper derivation, context, or justification. Several terms are undefined or poorly explained (e.g., who are $k$ and $\eta_t$ in Equation 1? What are $d_k$ and Norm in Equation 3?). The mathematical presentation lacks formalism, particularly in the sections dealing with attention mechanisms and regularization.

**W2**: The model introduces many components (kernel attention, G-LASSO, block-diagonal sampling, diffusion indices, etc.), which makes the architecture appear overly complex. Rather than a unified framework, the paper reads as a stack of connected techniques without clear empirical or theoretical motivation. For example, the sampling strategy based on retweets is described in detail but introduced without justification, theoretical grounding, or citations supporting its effectiveness or convergence.

**W3**: Baseline methods such as GCN, fully connected BERT, and neural networks without attention are not clearly introduced in the main text and appear abruptly: the paper does not provide any implementation details, such as hyperparameters or training protocols, for these baselines.

**Minor**: There are a few typos (e.g., "patters" or "medial" in the abstract).

---

### Review · Reviewer_kZR5 · 2025-06-01

**Summary Of Contributions:**

The paper introduces GRAD-T, a transformer that blends kernel-based temporal attention with a sparsity-regularized graph attention to model emotion diffusion across space and time. Two regularizers (1) an L1 penalty and (2) a sampling-based block-diagonal prior learn an interpretable latent influence network. Tests on a new COVID-19 Twitter dataset (10 M tweets, 879 U.S. locations) show that GRAD-T and its block-diagonal variant beat baselines (plain NN, GCN, fully connected transformer) on next-step emotion prediction and reveal that negative emotions (fear, anger) persist longer and spread farther than positive ones.

**Audience:**

Yes

**Broader Impact Concerns:**

The work leverages geolocated tweets, potentially exposing users’ emotional states and political attitudes at fine spatial resolution. This raises (i) privacy risks if raw data or embeddings can be de-anonymized, and (ii) manipulation concerns: marketers or political actors could exploit predicted diffusion pathways to target susceptible regions.

**Claims And Evidence:**

Yes

**Requested Changes:**

1. Expand baseline set: include state-of-the-art graph Transformers (Graphormer, SAN-based models) and temporal GNNs with positional encodings.
2. Full-scale evaluation: report results on the entire 879-location graph and at least one additional domain (e.g., retweet cascades or mobility).
3. Ablation study: isolate the effect of (a) kernel attention, (b) L1 regularisation, (c) BD sampling, (d) indices computation.
4. Statistical rigor: provide confidence intervals or paired tests over multiple random splits; add run-time and memory metrics.
5. Release artifacts: supply preprocessing scripts, hyper-param config files, and code for index visualizations to honour the reproducibility claim.
6. Clarify how theta and k are chosen in Algorithm 1; explore sensitivity.
7. Discuss limitations when retweet data are unavailable (how to seed the BD prior?).

**Strengths And Weaknesses:**

[Strengths]
- The paper is generally well structured.
- Novel fusion of transformer attention and graph sparsity; no predefined graph needed.
- Large real-world dataset, promised for public release, adds value.

[Weaknesses]
- No ablation disentangling the contribution of kernel vs. spatial regularization; limited theoretical insight into optimization properties of the BD heuristic.
- Lacks complexity analysis; training time and memory footprint on 879-node graph not reported; convergence behaviour of alternating optimisation unclear.
- Baselines omit strong recent graph Transformers (e.g., Graphormer, SAN-based GNNs); evaluation restricted to California subset, not full 879-node graph; no variance or statistical significance tests.
- Key preprocessing (BERT emotion classifier, topic model) is only sketched; no training scripts or random-seed control; block-diagonal sampling uses several heuristics (theta, k) without justification.
- Scalability and generalization to larger graphs or other domains are not shown.

[Minor issues]
- Typos (“higherst”); some symbols are undefined until later sections.

---

### Decision · Action_Editor_dv7B · 2025-07-16

**Recommendation:** Reject

**Audience:**

Yes

**Audience Explanation:**

The paper proposes a transformer that combines kernel-based temporal attention with a sparsity-regularized graph attention to model emotion diffusion across space and time. It may be of interest to a relatively small subset of TMLR's audience.

**Claims And Evidence:**

No

**Claims Explanation:**

The reviewers had a number of concerns regarding the presentation/clarity of the contribution as well as the experimental evaluation. However, the authors did not engage with the reviewers and addressed these concerns.